# Description of *Crinitomyces reliqui* gen. nov., sp. nov. and Reassignment of *Trichosporiella flavificans* and *Candida ghanaensis* to the Genus *Crinitomyces*

**DOI:** 10.3390/jof8030224

**Published:** 2022-02-24

**Authors:** Varunya Sakpuntoon, Gábor Péter, Marizeth Groenewald, Dénes Dlauchy, Savitree Limtong, Nantana Srisuk

**Affiliations:** 1Department of Microbiology, Faculty of Science, Kasetsart University, Bangkok 10900, Thailand; varunya.sa@ku.th (V.S.); fscistl@ku.ac.th (S.L.); 2National Collection of Agricultural and Industrial Microorganisms, Institute of Food Science and Technology, Hungarian University of Agriculture and Life Sciences, Somlói út 14-16, H-1118 Budapest, Hungary; peter.gabor@uni-mate.hu (G.P.); dlauchy.denes@uni-mate.hu (D.D.); 3Westerdijk Fungal Biodiversity Institute, Uppsalalaan, 8, 3584CT Utrecht, The Netherlands; m.groenewald@wi.knaw.nl; 4Biodiversity Center, Kasetsart University (BDCKU), Bangkok 10900, Thailand; 5Academy of Science, The Royal Society of Thailand, Bangkok 10300, Thailand

**Keywords:** ascomycetous yeast, *Crinitomyces flavificans* comb. nov., *Crinitomyces ghanaensis* comb. nov., four new taxa, *Crinitomyces reliqui* gen. nov., sp. nov.

## Abstract

The systematic position of 16 yeast strains isolated from Thailand, Hungary, The Netherlands, and the Republic of Poland were evaluated using morphological, physiological, and phylogenetic analyses. Based on the similarity of the D1/D2 domain of the LSU rRNA gene, the strains were assigned to two distinct species, *Trichosporiella flavificans* and representatives of a new yeast species. Phylogenetic analyses revealed that *Candida ghanaensis* CBS 8798^T^ showed a strong relationship with the aforementioned two species. The more fascinating issue is that *Candida* and *Trichosporiella* genera have been placed in different subphyla, Saccharomycotina and Pezizomycotina, respectively. The close relationship between *Trichosporiella flavificans*, *Candida ghanaensis* and the undescribed species was unexpected and needed to be clarified. As for morphological and physiological characteristics, the three yeast species shared a hairy colony appearance and an ability to assimilate 18 carbon sources. Based on phylogenetic analyses carried out in the present study, *Crinitomyces* gen. nov. was proposed to accommodate the new yeast species, *Crinitomyces reliqui* sp. nov. (Holotype: TBRC 15054, Isotypes: DMKU-FW23-23 and PYCC 9001). In addition, the two species *Trichosporiella flavificans* and *Candida ghanaensis* were reassigned to the genus *Crinitomyces* as, *Crinitomyces flavificans* (Type: CBS 760.79) comb. nov. and *Crinitomyces* *ghanaensis* (Type: CBS 8798) comb. nov., respectively.

## 1. Introduction

Several species concepts have been applied for yeast identification. A phenotypic species concept and growth profiles were initially used, while a biological species concept including data from mating experiments was later employed [1]. However, the phenotypic concept is limited due to the simplicity of fungal features, such as spore characters, which may lead to phenotypically cryptic taxa [2], and the lack of phenotypic divergence may also occur from the failure of accurate diagnosis [3,4,5]. With the introduction of sequencing technology, the sequence-based species concepts, including the phylogenetic analysis, became broadly applied and extensively employed in fungal taxonomy [1]. As a result, a more accurate classification of yeasts has been obtained.

A number of yeasts have been identified by phenotypic approach, often leading to misidentification or incorrect taxonomic assignment. One of these species-rich and polyphyletic yeast genera within Saccharomycetales order is the genus *Candida*. In the past, asexual yeasts with multilateral budding but showing nondistinctive cellular morphology were temporarily assigned to the genus *Candida* [6]. Currently, the placement of many *Candida* species is still unclear although sequencing technology has been frequently employed in yeast identification and classification. Phylogenetic analysis led to the recognition that various *Candida* species are distributed throughout the subphylum Saccharomycotina [7]. This situation is aggravated by the observation that even now some novel species are described in the *Candida* genus as a temporary placement based on short sequences and/or a small number of genes in phylogenetic analyses and, in addition, the lack of taxonomic characters needed to classify them elsewhere [8]. Fortunately, due to the availability of sufficient DNA sequence datasets, various *Candida* species have been transferred to new or already existing genera such as *Scheffersomyces* [9], *Danielozyma, Deakozyma, Middelhovenomyces* [10], *Diutina* [11], *Saturnispora* [12], *Groenewaldozyma* [13], *Teunomyces* [7] and *Limtongozyma* [14]. However, many *Candida* species are awaiting more analysis and accurate classification.

During our investigation of yeast communities in food waste, the strain DMKU-FW23-23 was found. The initial search of the GenBank using BLASTn search of the D1/D2 domain of the large subunit (LSU) of ribosomal RNA (rRNA) gene revealed that this yeast strain was distinct from the described yeast species in the database, but related with *Trichosporiella flavificans* CBS 760.79^T^ and *Candida ghanaensis* CBS 8798^T^. It is surprising that *Trichosporiella* and *Candida* are placed in different subphyla i.e., Pezizomycotina and Saccharomycotina, therefore it is unlikely that they are closely related to each other. Obtaining the correct placement and description of the new yeast species represented by strain DMKU-FW23-23 based on an integrative (polyphasic) taxonomic approach and reassignment of *Trichosporiella flavificans* and *Candida ghanaensis* were accomplished in this study.

## 2. Materials and Methods

### 2.1. Yeast Isolation

Table 1 presents the list of strains considered in this study. The strains from Thailand were isolated by the direct isolation method as described by Sakpuntoon et al. [15]. Yeast extract peptone dextrose (YPD) agar supplemented with 0.025% (*w*/*v*) sodium propionate and 0.02% (*w*/*v*) chloramphenicol was used as yeast isolation medium. The inoculated agar plates were incubated at 30 ± 2 °C until colonies appeared. Yeast colonies were selected based on different colony morphologies and then purified by cross streaking on YPD agar without antibiotics. The strains CBS 15,014 and CBS 142,641 were isolated from soil and sediment from wastewater treatment facility in The Netherlands, respectively, while the strain CBS 161.94 was isolated from sewage sludge in Katowice, the Republic of Poland. All Hungarian strains were isolated from Danube water. The water samples were taken from the surface of the river from the riverbank by sterile wide mouth screw capped bottles. The samples were kept in a refrigerator until they were processed within 24 h. Hundred milliliter aliquots of the samples were enriched in 500 mL yeast nitrogen base (YNB) medium supplemented with 0.5% (*v*/*v*) carbon-source (methanol or hexadecane) and incubated on a horizontal shaker for seven days (25 °C, 100 rpm), then 0.1 mL of each culture was transferred to a 16 mm culture tube containing 5 mL of liquid medium with the same composition. Following an additional week of incubation on a rotary shaker (25 °C, 30 rpm) the enriched cultures were serially diluted and surface plated on Rose-Bengal Chloramphenicol (RBC) agar. Representative strains were isolated on glucose (2%)-peptone-yeast extract (GPY) agar after 7 days of incubation at 25 °C in darkness and purified by repeated streaking.

For preservation of the isolated strains, a single yeast colony was cultured in a yeast extract malt extract (YM) broth for 18–24 h. Cell pellets were then collected by centrifugation, washed twice with sterile distilled water, and resuspended in fresh YM medium. The active yeast was then preserved in a metabolically inactive state by storing at −80 °C in YM broth supplemented with 30% (*v*/*v*) glycerol for long-term preservation.

### 2.2. DNA Sequencing and Phylogenetic Analysis

Yeasts were grown in YM broth for 18–24 h. The cell pellets were then collected and used for DNA extraction by enzymatic method [16]. The small subunit (SSU) rRNA gene, internal transcribed spacer (ITS) region and the D1/D2 domain of the large subunit (LSU) rRNA gene were amplified with the primer pairs SSU1f/SSU4r [17], SSU3f/SSU2r [17], NL5A/NS7A [18] and NL1/NL4 [19] respectively. The PCR products were purified with a FavorPrep^TM^ Gel/PCR Purification Mini Kit (Favorgen, Austria) and were then sent for DNA sequencing to First BASE Laboratories located at Seri Kembangan in Selangor state, Malaysia. Sequence assembly and alignment were conducted by the BioEdit version 7.0.5.3 program [20]. Aligned sequences were compared with the sequences in the GenBank database (http://www.ncbi.nlm.nih.gov/, accessed on 20 January 2022) using a BLASTn search. Phylogenetic trees were constructed based on the neighbor-joining method with the MEGA version 7.0.26 program [21]. Bootstrap analysis for the estimation of confidence levels of the clades was performed on 1000 bootstrap replications [22], and only values greater than 50% were shown. Table 2 shows the accession numbers of reference sequences retrieved from the GenBank database.

### 2.3. Phenotypic Characterization

The investigated yeasts were morphologically and physiologically characterized by standard methods described by Kurtzman et al. [23]. Yeasts were grown for three days in YM broth and YM agar at 25 °C for morphological study. Pseudo-hyphae and true hyphae formation were investigated on corn meal agar slide cultures at 25 °C for three days. Growth at different temperatures (15, 25, 30, 35, 37, 40, 42 and 45 °C) was determined in YM broth. The strains were examined individually or mixed in pairs for ascospore formation using different media including PDA, YM agar, YPD agar, corn meal agar, 5% malt extract agar, Gorodkowa agar, V8 agar, Fowell’s acetate agar [24] and yeast carbon base ammonium sulfate (YCBAS) agar [25] at 25 °C for up to twelve weeks with periodic microscopic inspection. Carbon and nitrogen source assimilation, carbohydrate fermentation, starch-like compounds production, and cycloheximide resistance tests were conducted in liquid media. A urea hydrolysis test was performed on a urea slant medium. Acid production and Diazonium Blue B (DBB) tests were conducted on solid medium in Petri dishes. All experiments were carried out with three replicates.

## 3. Results

### 3.1. Species Delineation and Molecular Phylogeny

BLASTn search analysis of the D1/D2 domain of the LSU rRNA gene against the GenBank database was performed to identify the yeast strain DMKU-FW23-23 found during a study of yeast community in food waste. The result showed that the top two results from a BLASTn search hit with the currently recognized species *Trichosporiella flavificans* CBS 760.79^T^ and *Candida ghanaensis* CBS 8798^T^, respectively. Surprisingly, these two species are described in different subphyla i.e., *T. flavificans* was placed in the subphylum Pezizomycotina and *C. ghanaensis* in the subphylum Saccharomycotina. The relationship between these two species was unexpected and needed to be clarified. Thus, a placement of the strains DMKU-FW23-23, *T. flavificans* CBS 760.79^T^ and *C. ghanaensis* CBS 8798^T^ was thoroughly investigated in this study. Seven additional strains, CBS 15240, CBS 15241, CBS 15242, CBS 15243, CBS 15014, CBS 161.94, and CBS 142641, that were similar to the strain DMKU-FW23-23, were found from BLASTn search analysis. Pairwise alignment revealed that the strain DMKU-FW23-23 and its companions differed from each other at 0–2 nucleotide substitutions without gaps in the D1/D2 domain of the LSU rRNA gene, while their ITS region showed no nucleotide substitutions and 0–1 gap (Table 3.).

All available strains with similar sequences to that of *T. flavificans* CBS 760.79^T^ in the GenBank database, DMKU-GTSC2-8, DMKU-GTSC2-2, DMKU-GTCC5-6, DMKU-GTCC5-12, DMKU-GTCC5-19, CBS 15244, and CBS 15245, were subjected to physiological and molecular analyses. The results of the pairwise alignment of *T. flavificans* CBS 760.79^T^ and its related strains are shown in Table 4. Identical sequences (0 nucleotide substitution with 0–1 gap) were found in the D1/D2 domain of the LSU rRNA gene and 0–3 nucleotide substitutions with 0–6 gaps were found in the ITS region among *T. flavificans* CBS 760.79^T^ and its related strains.

To find the accurate taxonomic placement of the strain DMKU-FW23-23 and its companions, *T. flavificans* CBS 760.79^T^ and *C. ghanaensis* CBS 8798^T^, a phylogenetic tree based on the D1/D2 domain of the LSU rRNA gene was constructed. In addition to the above-noted strains, related species were included in the analysis from the subphyla Saccharomycotina and Pezizomycotina. The results revealed that the strain DMKU-FW23-23 clustered with *T. flavificans* CBS 760.79^T^ and *C. ghanaensis* CBS 8798^T^ and their placements were found within the subphylum Saccharomycotina (Figure 1).

This result suggested that the assignment of *T. flavificans* to the genus *Trichosporiella*, which is nested in the subphylum Pezizomycotina, must be revised, and it should be transferred to the subphylum Saccharomycotina. In order to find its accurate placement within the subphylum Saccharomycotina, a phylogenetic tree based on a concatenated sequence of three genes including the small subunit (SSU) rRNA gene, the ITS region and the D1/D2 domain of the LSU rRNA gene was constructed (Figure 2 and Appendix A). The result demonstrated that the strain DMKU-FW23-23 and its companions formed a single lineage and were placed next to *T. flavificans* CBS 760.79^T^ and also grouped together with *C. ghanaensis* CBS 8798^T^. These three species formed a distinct monophyletic clade that is clearly separated from other described yeast species. Hence, a novel yeast genus namely *Crinitomyces* is proposed to accommodate *T. flavificans* and *C. ghanaensis* which are reassigned as *Crinitomyces flavificans* and *Crinitomyces ghanaensis*, respectively. Moreover, the strain DMKU-FW23-23 and its companion strains are also proposed as a novel yeast species within this novel genus and the name *Crinitomyces reliqui* sp. nov. is proposed.

Phenotypic characters of the three yeast species: *Crinitomyces*
*reliqui* sp. nov., *Crinitomyces flavificans* comb. nov. and *Crinitomyces ghanaensis* comb. nov. were compared and are summarized in Table 5.

A broad range of carbon sources was found to be assimilated, namely glucose, galactose, sorbose, cellobiose, maltose, sucrose, trehalose, melezitose, L-arabinose, D-xylose, erythritol, glucitol, mannitol, glycerol, ethanol, succinic acid, salicin and N-acetyl-glucosamine and only two carbon sources, inulin, and citric acid, were not assimilated by any of these yeasts. However, *Crinitomyces ghanaensis* CBS 8798^T^ did not show fermentation ability. It should be noted that, by all strains of *Crinitomyces flavificans*, a yellow pigment was exuded onto the agar medium during cultivation. For morphological characteristics, yeast colonies of the three species are white to cream, convex and butyrous, with a dull surface. All of the three yeast species showed a hairy colony morphology (Figure 3) which is the origin of the genus name, *Crinitomyces*.

### 3.2. Ecology

The type strain of *Crinitomyces*
*reliqui* DMKU-FW23-23^T^ was isolated from domestic food waste in Thailand while the two related strains, CBS 161.94 and CBS 142641, were isolated from similar types of habitats, sewage sludge and wastewater in Republic of Poland and The Netherlands, respectively. However, the strain CBS 15,014 was isolated from soil and the four related strains were isolated from water surface of a river in The Netherlands and Hungary, respectively. These habitats showed a contrast in terms of amount and type of nutrients available. The occurrence of this species in food waste, sewage sludge and rivers raised the possibility that the water of the rivers might have been polluted at the time of sampling. *Crinitomyces*
*reliqui* is suggested to be a cosmopolitan species because all eight strains of this species were found from four different countries.

The yeast *Crinitomyces flavificans* CBS 760.79^T^ was isolated from washings of ion-exchange resin in a guanosine monophosphate manufacturing plant in Japan whereas its seven companion strains were isolated from food waste and water in different countries. Therefore, *C. flavificans* should also be claimed as a cosmopolitan species.

The isolation sources and geographical origin of all investigated strains are summarized in Table 1.

### 3.3. Taxonomy

Description of *Crinitomyces* V. Sakpuntoon, G. Péter, M. Groenew., D. Dlauchy, S. Limtong & N. Srisuk, gen. nov.

MycoBank number: 842461.

*Crinitomyces* (Cri.ni.to.my’ces. N.L. fem. n. *Crinitomyces* refers to the hairy colony appearance of yeast within the genus).

Cells are spherical or ellipsoidal and asexual reproduction proceeds by multilateral budding. Septate hyphae are produced. Ascospore formation is not observed. DBB reaction is negative, starch-like compounds are not produced, and urea hydrolysis is negative.

Phylogenetic placement: Saccharomycetales, Saccharomycotina, Ascomycota.

Type species: *Crinitomyces flavificans* (Nakase) V. Sakpuntoon, G. Péter, M. Groenew., D. Dlauchy, S. Limtong & N. Srisuk comb. nov.

Description of *Crinitomyces*
*reliqui* V. Sakpuntoon, G. Péter, M. Groenew, D. Dlauchy, S. Limtong & N. Srisuk, sp. nov.

*Crinitomyces reliqui* (re.li.qu’i. L. fam. adj. *reliquum* of the residue; *reliquum* indicating that the type strain was isolated from residue of food or food waste).

After 3 days growth in YM broth at 25 °C, cells are spherical (1.5–3 μm) or ellipsoidal (1.5–2.0 × 2–3 μm). Colonies are white to cream, convex and butyrous, with a dull surface and filamentous margins (Figure 3c). True hyphae and branching lateral hyphae are observed on corn meal agar at 25 °C after 3 days (Figure 4b). Blasto-conidia are formed randomly from both hyphal types, globose to sub-globose, 1.8–3.0 μm. Ascospores are not found in individual cultures or in mixed cultures on PDA, YM agar, YPD agar, corn meal agar, 5% malt extract agar, Gorodkowa agar, Fowell’s acetate agar, V8 agar and YCBAS medium after 12 weeks at 25 °C. Glucose (viable), galactose (viable) and maltose (viable) are fermented. but lactose, sucrose, trehalose, melibiose and raffinose are not. Glucose, galactose, sorbose, cellobiose, lactose (weak), maltose (weak), melibiose (weak), sucrose (weak), trehalose, melezitose (weak), raffinose (weak), starch (variable), D-arabinose, L-arabinose, D-ribose, L-rhamnose, D-xylose, galactitol (weak), erythritol, D-glucitol (slow and weak), inositol (slow and weak), D-mannitol (slow and weak), glycerol, ribitol, ethanol, methanol (slow and weak), lactic acid, succinic acid, D-gluconic acid, α-Met-D-Glucoside, salicin (weak), N-Acetyl-D-Glucosamine, D-Glucono-5-lactone (weak), 2-Keto-D-gluconate (weak) and 5-Keto-D-gluconate (weak) are assimilated as the sole carbon sources, while inulin, citric acid, D-glucuronic acid and D-galacturonic acid are not assimilated. Ammonium sulfate, potassium nitrate (weak), sodium nitrite (weak), ethylamine hydrochloride, L-lysine and cadaverine dihydrochloride are utilized as sole nitrogen sources. Growth occurs in media containing 10% (*w*/*v*) sodium chloride/5% (*w*/*v*) glucose but not in 16% (*w*/*v*) sodium chloride/5% (*w*/*v*) glucose. Growth at 37 °C is positive for all strains except the strain CBS 15,014 of which growth occurs at 15 –30 °C. Growth is not observed in vitamin-free medium but variable results were found in medium supplemented with 0.01% (*w*/*v*) and 0.1% (*w/v*) cycloheximide. Acid production is variable. Urea hydrolysis, starch-like compounds production and diazonium blue B reaction are negative.

The holotype was isolated from domestic food waste in Bangkok, Thailand. The food waste sample was randomly collected via aseptic technique, and it was used for isolation process as previous described in materials and methods within 24 h. The obtained yeast colony was purified by cross-steaking on YM medium. After an initial BLASTn search analysis, additional representatives of the novel species were found. The strain was preserved at −80 °C in YM broth supplemented with 30% (*v*/*v*) glycerol. The holotype has been deposited and permanently preserved in a metabolically-inactive state in the Thailand Bioresource Research Centre (TBRC), Thailand, as TBRC 15054. An isotype has been permanently preserved in a metabolically-inactive state at the Department of Microbiology, Faculty of Science, Kasetsart University, Bangkok, Thailand as strain DMKU-FW23-23 and in the collection of the Portuguese Yeast Culture Collection (PYCC), Caparica, Portugal, as strain PYCC 9001. MycoBank number 842462.

New combinations.

*Crinitomyces flavificans* (Nakase) V. Sakpuntoon, G. Péter, M. Groenew., D. Dlauchy, S. Limtong & N. Srisuk, comb. nov.

MycoBank number: 842463.

Basionym: *Candida flavificans*, T. Nakase (1975). Antonie van Leeuwenhoek 41:202

Holotype: CBS 760.79, from washings of ion-exchange resin in a guanosine monophosphate manufacturing plant.

Note: *Crinitomyces flavificans* was first described as *Candida flavificans* based on an analysis of physiological and biochemical characteristics in 1975 [27]. Later, based on morphological characters it was reclassified as *Trichosporiella flavificans* [27]. However, the type species of *Trichosporiella, T. cerebriformis*, is nested in the subphylum Pezizomycotina. Based on the phylogenetic analyses carried out in this study, *T. flavificans* is reclassified here in Saccharomycotina as *Crinitomyces flavificans*. All phenotypic characters of the type strain, CBS 760.79, were re-examined in this study and results were found consistent with those of the first report. Nevertheless, some characters were differed among the companion strains and were then reported as “variable” as shown in the Table 5.

*Crinitomyces ghanaensis* (Kurtzman) V. Sakpuntoon, G. Péter, M. Groenew., D. Dlauchy, S. Limtong & N. Srisuk, comb. nov.

MycoBank number: 842464.

Basionym: *Candida ghanaensis*, C.P. Kurtzman (2001). Antonie van Leeuwenhoek 79:355.

Holotype: CBS 8798, from soil in Ghana.

## 4. Conclusions and Discussion

*Candida flavificans* CBS 760.79^T^ was isolated from washing of the ion-exchange resin in a guanosine monophosphate manufacturing plant and classified based on DNA base composition, proton-magnetic-resonance spectrum of polysaccharide, and serological characteristics [27]. However, in 1985, it was reclassified as *Trichosporiella flavificans* CBS 760.79^T^ due to a stronger coherence between hyphal cells and an absence of arthroconidia that split *Trichosporiella* from *Candida* [28]. Sequence analysis of this yeast genus has not been accomplished since then. In the era in which molecular study and sequencing technology play an important role in taxonomic study, yeast classification by the aforementioned analyses may not be sufficient and may also cause misidentifications.

Even if molecular methods and phylogenetic analyses have been used to identify yeast, unstable placement in the evolution line may occur, since they may be characterized based on short sequences and/or a small number of genes in phylogenetic analysis. *Candida ghanaensis* CBS 8798^T^ isolated from soil in Ghana, was first described with a weakly supported phylogenetic placement based on the D1/D2 domain of the LSU of rRNA gene by Kurtzman [26]. Subsequent analysis of the SSU of the rRNA gene sequence revealed that *C. ghanaensis* had a weak and probably insignificant affinity with *C**. incommunis*, but the highest matches in the GenBank database were found with members of the *Dipodascus**/Geotrichum* clade [8]. However, a well-supported placement of this yeast species has not yet established and a phylogenetic reconstruction from additional data was required.

Accurate classification of yeasts may not be achieved by a single method. Although molecular methods are irreplaceable, yeast identification requires combined application of several approaches (polyphasic taxonomy). Similarly, multiple conspecific strains are more reliable to propose a new yeast species. Nevertheless, it will also be more reliable if conspecific strains are isolated from different samples and/or geographical regions.

In this study we described a new yeast species, *Crinitomyces*
*reliqui*, which is closely related to *T. flavificans* and *C. ghanaensis* as they formed a well-supported clade in the phylogenetic trees and they also shared morphological and some physiological characteristics. We proposed here a novel yeast genus, *Crinitomyces*, to accommodate the novel species *Crinitomyces*
*reliqui* as well as *T. flavificans* and *C. ghanaensis*, which were reassigned as *Crinitomyces flavificans* and *Crinitomyces ghanaensis*, respectively.

## Figures and Tables

**Figure 1 jof-08-00224-f001:**
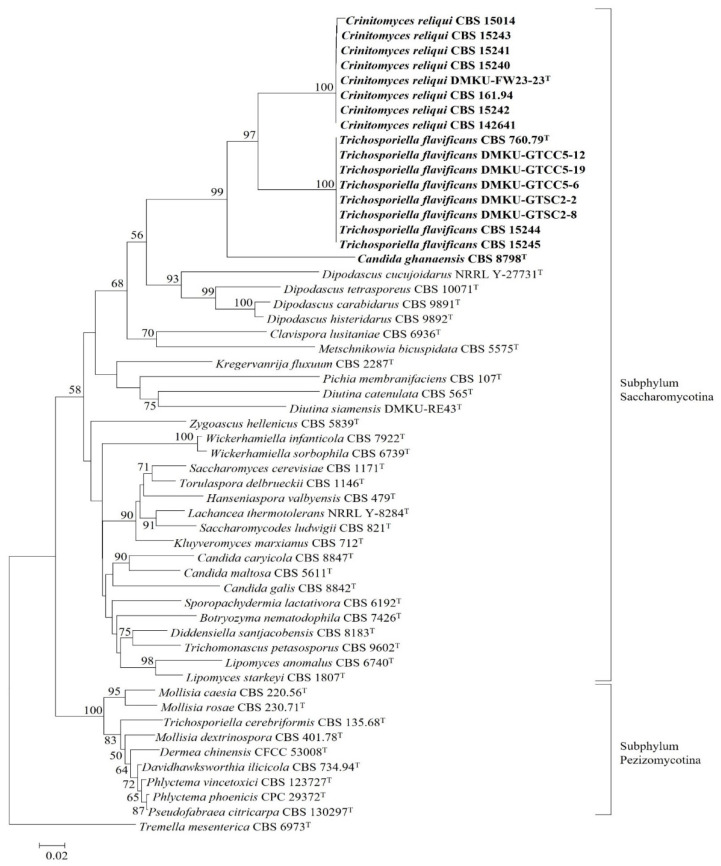
Phylogenetic tree based on the D1/D2 domain of the LSU rRNA gene showing an overview placement of *Trichosporiella flavificans*, *T. cerebriformis*, *Candida ghanaensis* and the novel species *Crinitomyces reliqui*. The phylogenetic tree was constructed using the neighbor-joining (NJ) method. The numbers provided on branches are frequencies with which a given branch appeared in 1000 bootstrap replications. Bootstrap values of less than 50% are not shown. *Tremella mesenterica* CBS 6973^T^ served as an outgroup species. Bar, 0.02 substitutions per site.

**Figure 2 jof-08-00224-f002:**
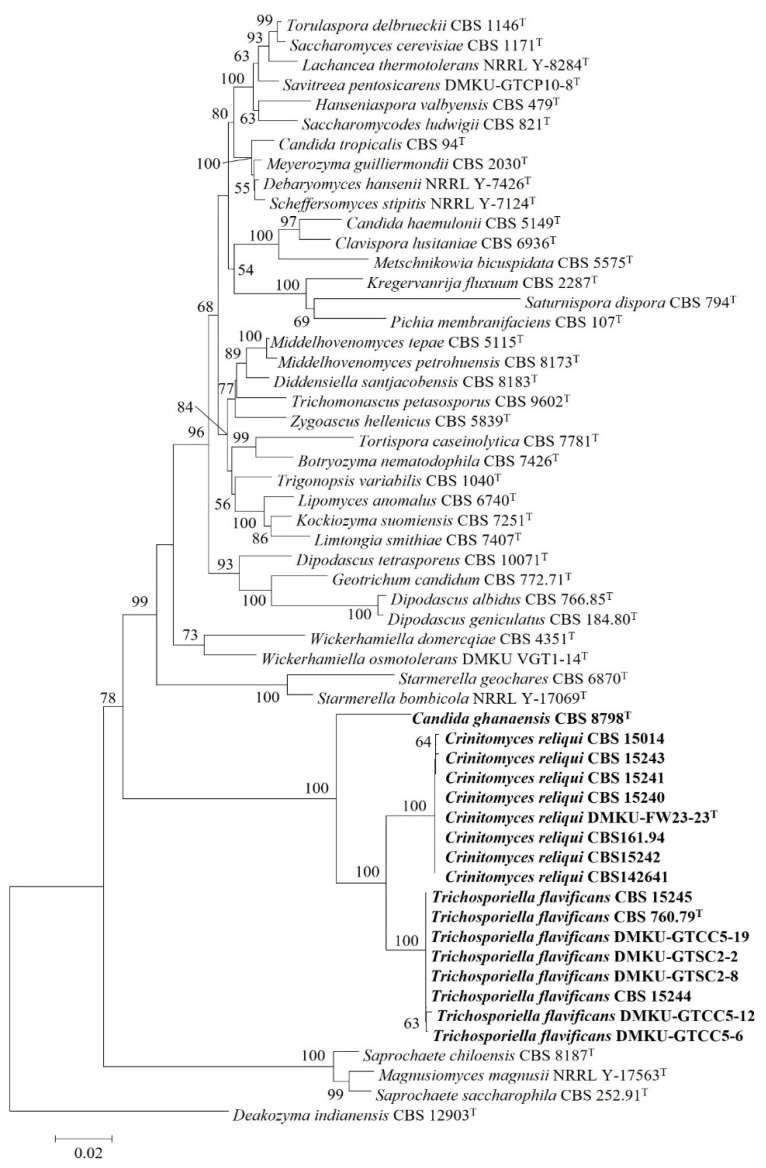
Phylogenetic tree based on concatenated sequences of the SSU rRNA gene, the ITS region and the D1/D2 domain of the LSU rRNA gene showing the placement of *Trichosporiella flavificans*, *Candida ghanaensis* and *Crinitomyces reliqui* within the subphylum Saccharomycotina. The phylogenetic tree was constructed using the neighbor-joining (NJ) method. The numbers provided on branches are frequencies with which a given branch appeared in 1000 bootstrap replications. Bootstrap values of less than 50% are not shown. *Deakozyma indianensis* CBS 12903^T^ served as an outgroup species. Bar, 0.02 substitutions per site.

**Figure 3 jof-08-00224-f003:**
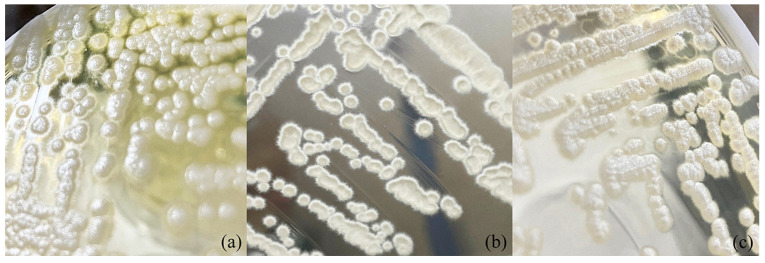
A hairy colony appearance of three yeast species on YM agar. (**a**) *Crinitomyces flavificans* CBS 760.79^T^; (**b**) *Crinitomyces ghanaensis* CBS 8798^T^ and (**c**) *Crinitomyces*
*reliqui* DMKU-FW23-23^T^ sp. nov.

**Figure 4 jof-08-00224-f004:**
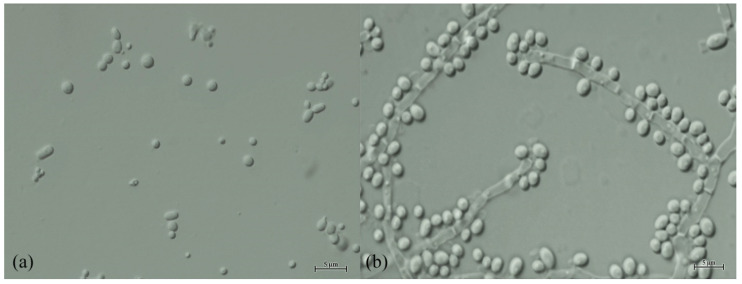
Morphology of *Crinitomyces*
*reliqui* DMKU-FW23-23^T^ (**a**) Cells in YM broth after 3 days at 25 °C (bar, 5 μm) and (**b**) True hypha formation on corn meal agar after incubated at 25 °C for 3 days (bar, 5 μm).

**Table 1 jof-08-00224-t001:** Yeast strains and isolation sources investigated in this study.

Yeast	Source of Isolation	Geographical Origin
*Crinitomyces**reliqui* sp. nov.		
DMKU-FW23-23^T^	Domestic food waste trap	Thailand
CBS 15014	Soil taken from 2 cm deep in Utrecht	The Netherlands
CBS 161.94	Sewage sludge in Katowice	Republic of Poland
CBS 142641	Sediment from wastewater treatment facility in Zeewolde	The Netherlands
CBS 15,240 (=NCAIM Y.01958)	Water of Danube Budapest (Location 1, 47.484163; 19.054271)	Hungary
CBS 15,241 (=NCAIM Y.02184)	Water of Danube Budapest (Location 2, 47.594721; 19.070331)	Hungary
CBS 15242	Water of Danube Budapest (Location 2, 47.594721; 19.070331)	Hungary
CBS 15,243 (=NCAIM Y.02185)	Water of Danube Budapest (Location 3, 47.592204; 19.069164)	Hungary
*Trichosporiella flavificans*		
CBS 760.79^T^	Washings of ion-exchange resin in a guanosine monophosphate manufacturing plant	Japan
DMKU-GTSC2-8	Food waste trap of Faculty of Science, KU canteen	Thailand
DMKU-GTSC2-2	Food waste trap of Faculty of Science, KU canteen	Thailand
DMKU-GTCC5-6	Food waste trap of KU central canteen	Thailand
DMKU-GTCC5-12	Food waste trap of KU central canteen	Thailand
DMKU-GTCC5-19	Food waste trap of KU central canteen	Thailand
CBS 15,244 (=NCAIM Y.02186)	Water of Danube Budapest (Location 3, 47.592204; 19.069164)	Hungary
CBS 15245	Water of Danube Budapest (Location 3, 47.592204; 19.069164)	Hungary
*Candida ghanaensis*		
CBS 8798^T^	Soil in Ghana	Ghana

KU: Kasetsart University, Bangkok.

**Table 2 jof-08-00224-t002:** The accession numbers of studied yeasts and reference sequences retrieved from the GenBank database.

Yeasts	Strain	SSU	ITS	D1/D2
*Botryozyma nematodophila*	CBS 7426^T^	NG061133	NR111167	NG042439
*Candida caryicola*	CBS 8847^T^	-	NR077194	NG055176
*Candida galis*	CBS 8842^T^	-	NR151797	NG058980
*Candida ghanaensis*	CBS 8798^T^	NG065532	KY102101	NG055180
*Candida haemulonii*	CBS 5149^T^	NG063413	NR130669	JX459759
*Candida maltosa*	CBS 5611^T^	-	NR138346	KY106554
*Candida tropicalis*	CBS 94^T^	EU348785	NR111250	NG054834
*Clavispora lusitaniae*	CBS 6936^T^	NG065595	NR130677	NG055408
*Crinitomyces* *reliqui*	DMKU-FW23-23^T^	OK275053	MW720560	OK298472
*Crinitomyces* *reliqui*	CBS 15014	OK275054	GU195664	OK298471
*Crinitomyces* *reliqui*	CBS 161.94	OK275055	MG250346	OK298463
*Crinitomyces* *reliqui*	CBS 142641	OK275056	MG250347	OK298464
*Crinitomyces* *reliqui*	CBS 15,240 (NCAIM Y.01958)	OK275057	MZ331539	OK298466
*Crinitomyces* *reliqui*	CBS 15,241 (NCAIM Y.02184)	OK275058	MZ312239	OK298468
*Crinitomyces* *reliqui*	CBS 15242	OK275059	MZ312240	OK298469
*Crinitomyces* *reliqui*	CBS 15,243 (NCAIM Y.02185)	OK275060	MZ312241	OK298470
*Davidhawksworthia ilicicola*	CBS 734.94^T^	-	NR154008	NG067307
*Deakozyma indianensis*	NRRL YB-1937^T^	NG061171	KJ476205	NG064315
*Debaryomyces hansenii*	NRRL Y-7426^T^	NG063420	NR120016	NG042634
*Dermea chinensis*	CFCC 53008^T^	-	NR171069	NG073667
*Diddensiella santjacobensis*	CBS 8183^T^	NG063433	NR151808	NG058985
*Dipodascus albidus*	CBS 766.85^T^	MK834548	AY788342	NG066154
*Dipodascus carabidarus*	CBS 9891^T^	-	NR155144	NG058292
*Dipodascus cucujoidarus*	NRRL Y-27731^T^	-	NR111352	NG055370
*Dipodascus geniculatus*	CBS 184.80^T^	NG064797	AY788301	NG066466
*Dipodascus histeridarus*	CBS 9892^T^	-	NR111351	NG042466
*Dipodascus tetrasporeus*	CBS 10071^T^	AB300502	AB300502	AB300502
*Diutina catenulata*	CBS 565^T^	-	NR077200	NG059158
*Diutina siamensis*	DMKU-RE43^T^	-	KT336715	KT336715
*Geotrichum candidum*	CBS 772.71^T^	JQ698930	HE663404	JQ689071
*Hanseniaspora valbyensis*	CBS 479^T^	NG063247	NR111113	NG042630
*Kluyveromyces marxianus*	CBS 712^T^	-	NR111251	NG042627
*Kockiozyma suomiensis*	CBS 7251^T^	NG062713	NR155335	NG055355
*Kregervanrija fluxuum*	CBS 2287^T^	NG063291	NR111196	NG042445
*Lachancea thermotolerans*	NRRL Y-8284^T^	NG061071	NR111334	NG042626
*Limtongia smithiae*	CBS 7407^T^	NG062712	NR138235	NG055354
*Lipomyces anomalus*	CBS 6740^T^	NG062697	KT923624	NG055345
*Lipomyces starkeyi*	CBS 1807^T^	-	NG055350	NR077109
*Magnusiomyces magnusii*	NRRL Y-17563^T^	MK834553	AY788307	MK834532
*Metschnikowia bicuspidata*	CBS 5575^T^	NG065596	KY104192	KY108455
*Meyerozyma guilliermondii*	CBS 2030^T^	NG063363	NR111247	NG042640
*Middelhovenomyces petrohuensis*	CBS 8173^T^	NG063431	NR156314	NG055211
*Middelhovenomyces tepae*	CBS 5115^T^	NG063435	NR154200	NG055181
*Mollisia caesia*	CBS 220.56^T^	-	MH857591	MT026503
*Mollisia dextrinospora*	CBS 401.78^T^	-	NR119489	MH872917
*Mollisia rosae*	CBS 230.71^T^	-	MH860088	MH871865
*Phlyctema phoenicis*	CPC 29372^T^	-	NR155690	NG067319
*Phlyctema vincetoxici*	CBS 123727^T^	-	NR145310	NG067282
*Pichia membranifaciens*	CBS 107^T^	NG064813	NR111195	NG042444
*Pseudofabraea citricarpa*	CBS 130297^T^	-	NR154319	NG069282
*Saccharomyces cerevisiae*	CBS 1171^T^	NG063315	NR111007	NG042623
*Saccharomycodes ludwigii*	CBS 821^T^	NG063254	NR165986	NG042629
*Saprochaete chiloensis*	CBS 8187^T^	NG070306	AY788349	MK834538
*Saprochaete saccharophila*	CBS 252.91^T^	NG070310	AY788309	MK834545
*Saturnispora dispora*	CBS 794^T^	EF550358	NR155832	NG055103
*Savitreea pentosicarens*	DMKU-GTCP10-8^T^	NG073529	NR172171	NG073813
*Scheffersomyces stipitis*	NRRL Y-7124^T^	NG063362	NR165947	NG042637
*Sporopachydermia lactativora*	CBS 6192^T^	-	NR111310	KY109772
*Starmerella bombicola*	NRRL Y-17069^T^	JQ698924	NR121483	NG042648
*Starmerella geochares*	CBS 6870^T^	NG065473	KJ630497	NG060806
*Tortispora caseinolytica*	CBS 7781^T^	NG065577	NR154482	NG055343
*Torulaspora delbrueckii*	CBS 1146^T^	NG061300	NR111257	NG058413
*Tremella mesenterica*	CBS 6973^T^	-	NR155937	NG069419
*Trichomonascus petasosporus*	CBS 9602^T^	NG062797	NR155940	NG055332
*Trichosporiella cerebriformis*	CBS 135.68^T^	-	NR155940	MH859089
*Trichosporiella flavificans*	CBS 760.79^T^	OK275050	MH873011	OK298462
*Trichosporiella flavificans*	DMKU-GTSC2-8	OK275046	MN460331	OK283398
*Trichosporiella flavificans*	DMKU-GTSC2-2	OK275045	MN460330	OK283396
*Trichosporiella flavificans*	DMKU-GTCC5-6	OK275047	MN460342	OK283393
*Trichosporiella flavificans*	DMKU-GTCC5-12	OK275048	MN460340	OK283395
*Trichosporiella flavificans*	DMKU-GTCC5-19	OK275049	MN460339	OK283397
*Trichosporiella flavificans*	CBS 15,244 (NCAIM Y.02186)	OK275052	MG250348	OK298465
*Trichosporiella flavificans*	CBS 15245	OK275051	MZ331540	OK298467
*Trigonopsis variabilis*	CBS 1040^T^	NG061132	NR154506	NG055341
*Wickerhamiella domercqiae*	CBS 4351^T^	NG061104	DQ911462	NG055328
*Wickerhamiella infanticola*	CBS 7922^T^	-	NR155985	NG058278
*Wickerhamiella osmotolerans*	DMKU VGT1-14^T^	MN192121	MN194615	MH141490
*Wickerhamiella sorbophila*	CBS 6739^T^	-	NR155987	NG055325
*Zygoascus hellenicus*	CBS 5839^T^	NG063434	NR111258	AY447007

^T^: Type strain of species.

**Table 3 jof-08-00224-t003:** Pairwise DNA sequence comparisons between the strain DMKU-FW23-23 and its related strains.

Yeasts	Nucleotide Substitution (bp)/Gap (bp)/Percentage of Sequence Similarity (%)		
	SSU	ITS	D1/D2
CBS 15,240 (NCAIM Y.01958)	2/0/99.9	0/0/100	0/0/100
CBS 15,241 (NCAIM Y.02184)	0/0/100	0/0/100	0/0/100
CBS 15,243 (NCAIM Y.02185)	0/0/100	0/0/100	0/0/100
CBS 15242	0/0/100	0/1/99.9	0/0/100
CBS 15014	0/0/100	0/1/99.9	2/0/99.6
CBS 161.94	0/0/100	0/0/100	0/0/100
CBS 142641	0/0/100	0/1/99.9	0/0/100

**Table 4 jof-08-00224-t004:** Pairwise DNA sequence comparisons between *Trichosporiella flavificans* CBS 760.79^T^ and its related strains.

Yeasts	Nucleotide Substitutions (bp)/Gaps (bp)/Percentage of Sequence Similarity (%)		
	SSU	ITS	D1/D2
*T. flavificans* DMKU-GTSC2-8	0/0/100	0/0/100	0/0/100
*T. flavificans* DMKU-GTSC2-2	1/0/99.9	1/0/99.8	0/0/100
*T. flavificans* DMKU-GTCC5-6	1/0/99.9	1/1/99.8	0/1/99.8
*T. flavificans* DMKU-GTCC5-12	0/0/100	3/6/98.6	0/0/100
*T. flavificans* DMKU-GTCC5-19	0/0/100	1/2/99.8	0/1/99.8
*T*. *flavificans* CBS 15,244 (NCAIM Y.02186)	0/0/100	0/0/100	0/0/100
*T. flavificans* CBS 15245	0/0/100	1/1/99.9	0/0/100

**Table 5 jof-08-00224-t005:** Physiological characteristics of *Crinitomyces*
*reliqui* in comparison to its closely related species.

Physiological Characteristics	1	2	3
Fermentation
Glucose	v	+	-
Galactose	v	v	-
Lactose	-	-	-
Maltose	v	-	-
Melibiose	-	-	-
Sucrose	-	-	-
Trehalose	-	v	-
Raffinose	-	-	-
Carbon assimilation
D-glucose	+	+	+
Galactose	+	+	+
Sorbose	+	v	v
Cellobiose	+	+	+
Lactose	w	+	-
Maltose	w	+	+
Melibiose	w	-	-
Sucrose	w	v	+
Trehalose	+	+	+
Melezitose	w	v	+
Raffinose	w	v	-
Inulin	-	-	-
Starch	v	v	-
D-arabinose	+	v	-
L-arabinose	+	+	+
D-ribose	+	+	-
L-rhamnose	+	-	-
D-xylose	+	+	+
Galactitol	w	-	-
Erythritol	+	+	+
D-glucitol	s/w	v	+
Inositol	s/w	-	-
D-mannitol	s/w	v	+
Glycerol	+	+	+
Ribitol	+	v	-
Ethanol	+	s/w	+
Methanol	s/w	s/w	-
Citric acid	-	-	-
Lactic acid	+	+	-
Succinic acid	+	+	+
D-gluconic acid	+	v	-
D-glucuronic acid	-	-	nd
D-galacturonic acid	-	-	nd
α-Met-D-glucoside	+	-	+
Salicin	w	+	+
N-acetyl-D-glucosamine	+	+	+
D-Glucono-5-lactone	w	+	nd
2-Keto-D-gluconate	w	v	-
5-Keto-D-gluconate	w	-	-
Nitrogen assimilation
(NH_4_)_2_SO_4_	+	+	nd
KNO_3_	w	v	-
NaNO_2_	w	w	nd
Ethylamine-HCl	+	+	nd
L-lysine	+	+	nd
Cadaverine	+	+	+
Others
Diazonium Blue B	-	-	nd
Starch-like compounds	-	-	-
Growth at vitamin free medium	-	-	-
Urea hydrolysis	-	-	nd
Growth at 15 °C	+	+	nd
Growth at 25 °C	+	+	+
Growth at 30 °C	+	+	+
Growth at 35 °C	v	+	+
Growth at 37 °C	v	+	+
Growth at 40 °C	-	+	-
Growth at 42 °C	-	-	-
Growth at 45 °C	-	-	-
0.1% Cycloheximide resistance	v	-	+
0.01% Cycloheximide resistance	v	-	+
Growth in medium supplemented with 50% glucose	+	v	nd
Growth in medium supplemented with 60% glucose	+	+	nd
Growth in medium supplemented with 5% glucose +10%NaCl	+	+	+
Growth in medium supplemented with 5% glucose +16%NaCl	-	-	nd
Acid production	v	v	nd

Species: 1, *Crinitomyces reliqui* sp. nov. (DMKU-FW23-23^T^ and seven additional strains); 2, *Crinitomyces flavificans* comb. nov. (CBS 760.79^T^ and seven additional strains) and 3, *Crinitomyces ghanaensis* comb. nov. CBS 8798^T^ [26] and data obtained from CBS. Abbreviation: +, positive; -, negative; s, slow positive; w, weak positive; v, variable (some strains are positive, others negative); nd, no data.

## Data Availability

All sequence data are available in NCBI GenBank following the accession numbers in the manuscript.

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
