# Peer review of "Description of Crinitomyces reliqui gen. nov., sp. nov. and Reassignment of Trichosporiella flavificans and Candida ghanaensis to the Genus Crinitomyces"

_jof, 2022, doi:10.3390/jof8030224_

Round 1
Reviewer 1 Report
The submitted manuscript focuses on the morphological and physiological properties as well as phylogenetic analyses of the group of 16 yeast strains, which included the strains isolated from eight sources in six countries. Based on previous studies, some of the strains were placed in the two species of the distinct subphyla – Trichosporiella flavificans (subphylum Pezizomycotina) and Candida ghanaensis (subphylum Saccharomycotina). The authors have recognized relationship between the two species, accommodated them to the newly proposed genus Crinitomyces, and showed their correct placement to the subphylum Saccharomycotina. Moreover, among the strains inspected, a group of strains formed monophyletic clade with the strains of the two species, but a single lineage next to T. flavificans and, therefore, the authors described a new species Crinitomyces reliqui.
The topic is an important subject. The manuscript is well written, clearly stated and well-organized. It contains interesting results on the diversity, properties and systematic position of the yeast strains studied.
I recommend this manuscript for publication after a minor revision as follows:
Page 1, line 35: Several species concepts have been applied for yeast species delineation. Please, reconsider the sentence …applied for the yeast identification?
Page 1, lines 37-39: However, the phenotypic characteristics… Please, rewrite this sentence, it is not clear.
Page 1, lines 43-44: However, there is no simple… I recommend to omit this sentence.
Author Response
Response to Reviewer 1
Thank you for your kind consideration and review. I would like to response to your comments as follows:
Page 1, line 35: Several species concepts have been applied for yeast species delineation. Please, reconsider the sentence …applied for the yeast identification?
Response: We edited the sentence to “Several species concepts have been applied for the yeast identification” as you suggested.
Page 1, lines 37-39: However, the phenotypic characteristics… Please, rewrite this sentence, it is not clear.
Response: We rewrite the sentence as “However, the phenotypic concept is limited due to the simplicity of fungal features, such as spore characters, may lead to phenotypically cryptic taxa [2], and the lack of phenotypic divergence may also occur from the failure of accurate diagnosis [3-5].”
Page 1, lines 43-44: However, there is no simple… I recommend to omit this sentence.
Response: We deleted this sentence as you suggested.
Reviewer 2 Report
I enjoyed reading this paper. It happens very rarely that I have no suggestions for improvement. Congratulations to the authors.
The authors may decide to add a ML tree to their results -- with these multi-locus data in hand, I probably would have done a ML analysis using models for each partition (5 partitions: SSU, ITS1, 5.8S, ITS2, LSU) in IQ-TREE, as a best practice approach. I personally only use MEGA for quick and dirty first assessments of evolutionary relationships, which I then optimize by using model based approaches in programs optimized for such analyses (e.g., IQ-TREE for partitioned ML).
Author Response
Response to Reviewer 2:
Thank you for your kind consideration and review. I would like to response to your comments as follows:
We decided to add Figure S1 which is a ML tree constructed using the IQ-Tree for partitioned ML as you suggested and the result showed the consistent outcome with Fig. 2 which the phylogenetic tree was constructed using the neighbor-joining (NJ) method.
